# Learners’ Knowledge of Environmental Education in Selected Primary Schools of the Tshwane North District, Gauteng Province

**DOI:** 10.3390/ijerph192315552

**Published:** 2022-11-23

**Authors:** Ernest Khalabai Mashaba, Simeon Maile, Malose Jack Manaka

**Affiliations:** Department of Educational Foundation, School of Education, Soshanguve North Campus, Tshwane University of Technology, Pretoria 0183, South Africa

**Keywords:** environmental education, knowledge, conservation, clean environment, curriculum

## Abstract

This study focused on determining the level of environmental education knowledge and skills among primary school learners in South Africa, under the National Curriculum Statement Grades R-12. Environmental education is important in the promotion of sustainable environmental conservation and reducing negative effects, which are caused by adverse climatic conditions. We used a qualitative approach for data collection from eight primary schools located in the North Tshwane District of Gauteng province, through interviews and observations. The sample comprise teachers, learners, and non-teaching staff working at the schools. The findings from interviews suggest that the curriculum offers learners adequate environmental educational knowledge in the Social Science and Natural Science subjects. Similarly, the results from the observations show that, from visual sights of the surroundings of the schools, the school has gardens, tree nurseries, tree plantations, and a clean environment around the school. In conclusion, learners are provided with adequate environmental educational knowledge and they are able to contribute towards maintaining a clean environment and conservation in their communities.

## 1. Introduction

The development of children requires that they are provided with relevant education and skills to enable them to understand what challenges exist in society, and what they can do within their capacity to contribute to sustainability in their society, especially in fighting climate change. Primary school level is the first point of entry in the education sector, where learners come into contact with world education. Therefore, this entry point provides an opportunity for education stakeholders to ensure that they design and provide a curriculum for teaching that incorporates environmental education (Kimaryo, 2011) [1].

As per the 2020 national statistics, South Africa had a population of 58.7 million in 2019. The birth rate was at 21 births per 1000 people, with a death rate of nine deaths per 1000 people. By average, the annual growth rate is 1.43%. This growth is alarming, as it is causing gradual pressure on the natural environment. There is a possibility that, if necessary actions are not taken, the region will suffer from the depletion of resources and possible arid conditions will set in. This aspect calls for the need to conserve the environment and sensitize the population on conservation importance and how they can participate in conservation efforts. Children at the primary school level are in a good position to be provided with environmental education so that they can grow, while being well informed and ready to embrace conservation efforts.

Environmental learning focuses, either as a theme or principle, is an unprecedented change that has been introduced to the South African curriculum, and this calls for stakeholders, such as teachers, policy makers, parents, and learners, to embrace the knowledge and ensure that every individual in the society are well informed and working to ensure that everybody is well informed regarding society’s conservation efforts (Le Grange and Reddy, 1997) [2].

Similar to many other countries, South Africa is one of the members of the International Conferences on the Environment. The country has responded to global concerns on the environment and international declarations by including Environmental Education (EE) to the curriculum, which has been taught in schools since the late 1990s. This initiative by SA government to integrate EE to the education curriculum is commendable. Conservation is an ideal concept that should be embraced, as it helps to reduce adverse climate change in our environment. Moreover, the economy of South Africa relies heavily on the country’s natural resources, such as vegetation and minerals. Therefore, there is need for the country to take conservation seriously to safeguard the environment for future benefits to be achieved (Sullivan, 2013) [3].

The life support of the environment is being threatened by manmade environmental issues and problems such as floods, draughts, poor sanitation, and land degradation because of engaging in poor agricultural practices, unsuitable mineral harvesting practices, and loss of biodiversity. Despite some of these activities increasing the food basket of the country, they need to be controlled in a manner that will not pollute the environment and affect future sustainability. Educational activists in South Africa should be made aware that there is a growing challenge and urgent need to reconceptualize the process of listing important aspects of conservation in the environment. This will ensure that they are able to sensitize other stakeholders in the education sector to take conservation seriously to save the world from the degradation of the environment, which directly contributes to climate change.

This research was conducted with the aim to determine the level of environmental education awareness that the education sector is contributing to learners at the primary school level. Understanding environmental conservation from the early development stage of primary school provides learners with an opportunity to understand and develop knowledge on the importance of conserving the environment. The objectives, therefore, are:Determine whether learners are provided with adequate environmental education at the primary school level.Analyse conservation efforts that are undertaken by learners, which are related to the learning outcomes of environmental education provided in South African primary schools.Identify challenges and opportunities that are facing environmental education provision on primary school education level in South Africa.

Despite the incorporation of environmental education into the curriculum, research has noted that schools face challenges of inadequate resources and a varying number of staff members, both in teaching and teaching staff. This is a hindrance to policy and environmental education teaching and implementation, because resources are needed to reduce the learner to teacher ratio when teaching, to enable learners to obtain adequate attention from their teachers. This aspect will contribute to increase learners’ understanding of what conservation is and how they can adequately control the degradation of the environment. The fact that South Africa is a developing country contributes to the challenge of lack of enough resources to finance the education sector. The challenge of accessing resources has made it difficult for a balanced development in terms of infrastructure, especially in public learning facilities such as primary schools that rely on government funding to finance their operations. The ministry of education has more to do in soliciting more resources so that it can be channeled to learning centers to learners provided with sustainable infrastructure, learning materials such as books, and hiring of teachers who will be used to provide teaching to learners.

The implementation of EE in the education system is not an easy concept, and this has provided a challenge to educators. Esland (1971) [4] noted that the introduction of EE into the school curriculum represents challenges to the dominant concepts of organization and information transmission. This aspect causes conflict among teachers with their teaching and learning approach. Therefore, the integration of EE into the education system, and the implementation of this knowledge has been a challenge that made learners unable to adequately access and practice what EE information entails.

## 2. Literature Review

The motive behind teaching environmental knowledge at the primary school level is due to the need to develop future leaders who are able to make informed decisions and contribute positively towards the overall development of society. To teach effectively, teachers should have adequate knowledge on the content that they are required to teach. This implies that, during the training of teachers before they are deployed, they should have practical and relevant skills on environmental conservation to convey the needed information to allow learners to understand and practice conservation activities. Shulamn (1986) [5] emphasized the need for teachers to be adequately educated so that they can teach learners important concepts, especially on conservation and climate change. According to Shulman (1986), [5] teacher’s knowledge, which is referred to as pedagogical context knowledge, should contain the concept that the teacher is going to teach to the learners in the school.

EE contemporary forms were first introduced in South Africa in the mid-1970s by the Belgade Charter of 1975 and Tbilisi principles of 1977 as analyzed by Gough (2006) [6]. Before the introduction of this information, prior knowledge only focused on basic ecology interpretation and soil erosion discussions, which were not detailed. As part of the government intervention to ensure that EE was embraced and being taken seriously in the conservation of the environment, the African National Congress embraced conservation ideas and it was implemented in the 1994 education framework. This led to the establishment of the EE curriculum initiative as a state and civil society partnership to help in curriculum development to design the C2005 curriculum.

The inception of C2005 enabled themes such as environmental management scope to be taught in schools. Concepts such as environmental management and all forms of pollution were embedded in the curriculum, and this led to further emphasis on conservation and environmental management. Issues such as pollution, deforestation, and overpopulation were also prioritized in the curriculum so that the learners could be adequately sensitized on the importance of conserving the environment for sustainable growth and development.

The main aim of including EE is to align global education with environmental concerns that is rapidly rising worldwide (Monroe, Oxarart and Plate, 2013) [7]. The protection of the environment is entirely the responsibility of each individual in society, because every individual is living in this society, and they must take care of the environment where they are residing (Jorgenson, Stephens and White, 2019) [8]. The millennium development goal of creating environmental sustainability provides an opportunity for stakeholders in society to draft an appropriate framework, which can be used to safeguard the environment and protect it from any form of degradation.

By virtue of education provision, the education sector is in a good position to transform society by providing knowledge and skills. Separating the education system into distinct levels, primary school, secondary, and tertiary education levels, provide an opportunity to stakeholders to design a curriculum that is suitable, based on the thinking capabilities of the learners in each education level. Learners in the primary school level are in a better position to be provided with EE gradually, because this is a stage in which they are learning and being introduced to general knowledge and technical skills that enable them to grow and develop positive knowledge, as they develop on matters of conservation in the society. The fact that many of the primary schools in South Africa are located in the rural and peri-urban areas provide an opportunity for the education sector to design the curriculum in a way that will enable learners to embrace conservation efforts from their early development years (Stanišić and Maksić, 2014) [9].

To achieve the millennium development goals, illiteracy should be eradicated in society, and this can be achieved through empowering and adequately financing the education sector, thus allowing stakeholders in the sector to educate the general public on the importance of environmental conservation. This is essential in ensuring that future generations are well equipped with environmental conservation information, and they can adequately relate and take part in conservation efforts. Stanišić and Maksić (2014) [9] claimed that environmental education is the process of developing awareness among the people and enable people to acquire knowledge on the environment, how they can conserve it, what negative effects arise from neglecting and polluting the environment, and what does it take to encourage and foster sustainability of the ecosystem. According to Nhamo and Mjimba (2020) [10] EE intends to:Help in improvement of the quality of environment.Create awareness on the challenges and problems affecting the society, which are related to neglecting of the environment.Create an ideal and enabling environment for people to make and enforce conditions related to conservation, and how they can conserve the environment.Foster development of mechanisms, which can be used to execute environmental development plans.

Environmental challenges, such as conservation, pollution, population growth, and scarcity of resources, are all important aspects, which should be translated into curricular content that can easily be taught in education institutions (Kováčová and Vacková, 2014) [11]. The curriculum should be reviewed and changed periodically to ensure that any development in the environment is incorporated, so that learners are aware of it and can participate in transformation of the environment. Whereas environmental education can be specific to a certain geographical location, many issues within the broad concept of environment are similar globally, and this broadly categorize EE into three distinct groups namely biological, physical, and socio-cultural (Erdoğan, Kostova and Marcinkowski, 2009) [12].

Curriculum development is an all-round initiative that is influenced by a number of factors, such as political, economic, social, and technological. With regard to the economic contribution, the economic development of a country depends on education development. During curriculum development, the developers must ensure that they design a curriculum that promotes economic development in the country (Dube and Jita, 2018) [13]. The type of knowledge that learners receive in education institutions will enable them to understand and contribute towards economic development and, therefore, the designers of the curriculum must ensure that it incorporates every aspect that can lead to development in society.

In the 21st century, technology commands and control a lot of activities and operations, including learning institutions. Curriculum developers must ensure that the type of curriculum that they design has room to incorporate technology so that learners are able to learn relevant development and transformation aspects, which will result in even development of the society. Incorporating technology in curriculum development will therefore empower learners to study basic education skills and understand what technology entails and how they can tap into growing technology to expand their knowledge, talents, skills, and knowhow (Molapo and Pillay, 2018) [14].

The political situation of the country also influences curriculum development. The type of leadership and transformation that are being undertaken by the elite class also influence the outcome of curriculum development, especially in designing the history subject curriculum. Various political transformations that have taken place in the country must be incorporated into the curriculum so that learners can understand transformations from the colonial era to the present, and how transformation helped in improving the life of South Africans. On the other hand, the cultural aspect revolves around community development, origin, and interactions. This concept must be incorporated into the curriculum so that the next generation can understand their origin and what it takes to conserve their traditions.

The rising global warming effect, and the eventual rising of water levels in the oceans due to melting of polar ice, call for curriculum developers to work towards ensuring that development in curriculum goes hand in hand with what is currently happening in society, so that the millennials and Generation Z can understand what is needed to control such adverse climate change conditions. Therefore, the curriculum should reflect and address the multiple challenges that affect society.

The future development phases of the curriculum should incorporate issues such as gardening, littering, environmental management, and vandalism, as well as food production, pollution, natural disasters, and recycling activities. The education development, as noted by Simpson (2002) [15], should incorporate pertinent issues affecting society, and which are developing. Collins (1980) [16] noted that EE must be provided extensively and cover issues that are affecting societal development, especially matters to do with global warming and climate change.

Learning cannot be undertaken without access to relevant learning materials such as books and other publications, which are used as reference materials for teaching. Educators have been at the forefront of the challenging and changing situations in which we have found ourselves, whether politically, psychologically, emotionally, or physically. Teachers are game changers and need to be well-equipped to change the world. Their readiness means a lot to us. Education has always been the most functional tool for tackling problems, and there is no doubt that today’s education will influence the world of tomorrow. Therefore, education can be one of the most important tools for addressing the challenges of human advancement in the contemporary world (Bogner and Wiseman, 1997; Rezaei et al., 2016) [17]. EE development is one of the main tasks of any community (Heimlich and Ardoin, 2008) [18]. Therefore, the transfer of EE to students is the task of schools. In this regard, teachers should be prepared to transfer knowledge and environmental information to students [19]. The education sector therefore requires the active transformation and periodic review of learning materials in order to ensure that the curriculum contains secondary points of materials for reference.

## 3. Research Strategy

We collected data through a qualitative approach from teachers, learners, and non-teaching staff working in primary schools located in the North Tshwane District, South Africa. The primary purpose of this investigation was to determine the level of understanding of EE among learners. We framed our research within the interpretive paradigm, which emphasizes the exploration of meaning through language and understanding of the context in which events occur (Bailey, 2014) [20]. The sample of the study included teachers (*n* = 10), non-teaching staff (*n* = 8), and learners (*n* = 80).

We used interviews and observations techniques to collect the data. Large amounts of information were collected through the interviews. With the observation technique, we observed learner behavior and attitudes within specific formal and informal contexts, to obtain accurate information relating to the environment and environmental education as a whole. The main reason why we used observations is that an environment is the perfect reflection of attitudes and knowledge, even when people are absent according to Van and Dunlap (1981) [21]. Scientists use observation to collect and record data (Eberbach and Crowley, 2009) [22]. We used standardized open-ended questions (Turner, 2010) [23]. We kept the questions identical throughout the interviews (Aksu, 2009) [24].

We analyzed the data through the use of a content analysis technique where we read the content and grouped it into themes, which can be easily analyzed and interpreted. After coding of the research, we were able to generate and understand concepts that were collected from the research, based on the interpretation and thoughts of participants that we engaged in the study. Because the research involved three different groups of participants, there was a need to expand the findings to understand what is contained in the information collected, and how most of the participants answered the questions presented to them. The technique involved the reformulation of stories presented by the participants, taking their thoughts into account, and analyzing different experiences of participants to generate comprehensive research findings that correlate with environmental knowledge research needs.

## 4. Findings

### 4.1. Teaching Environmental Education as a Curriculum Subject 

The participants stated that the main subjects, which helped them to learn about environmental issues by providing them with an opportunity to teach and learn, respectively, about the environment, were *natural science* and *social science*. They noted that these two subjects contain various provisions that helped in shaping learners’ access to basic knowledge on environmental studies. One teacher said: “*since the introduction of the new curriculum, EE is catered for. In natural science and social science there is a provision for EE*”. Earth education is perceived by the teachers interviewed as a competent and necessary EE, which will help learners to build a strong relationship with the natural world and enable them to interact with the living world. Another teacher said: “*earth education topic is very important in that it raises the dangers of climate change and urges us to be conscious of how use natural resources*”. Teachers noted that providing EE studies through social science subjects encouraged the development of sensory awareness among learners, which help them to participate in environmental conservation efforts by developing an understanding of life, and what contribution they can make towards sustainability and green conservation efforts. One learner concurs with teachers and said: “*I did not know that the earth is in trouble from human footprint. I like what our natural science and social teachers are teaching us. We need to preserve the world for the future*”.

Social science studies in South Africa covers History and Geography. The content of these subjects relates to the natural ecosystem and environmental impact of human activities towards the ecosystem. As noted in research by DoE (2021) [25], this curriculum teaches learners issues related to the environment by incorporating aspects such as significance of water bodies and how they can be conserved, and human activities and how they contribute towards conservation of a natural ecosystem.

Natural science, as noted by the teachers who were interviewed, combines life science and chemical and physical sciences. The participants noted that the natural science subject accounted for 50% of environmental issues in the form of the contribution of natural science in conservation and preservation of the environment. On the use of materials for science projects, another learner said, “*we are taught that some of the chemicals are dangerous and can destroy flora and fauna. We learn that such toxic materials must be disposed at a prescribed space under the supervision of adults*”. The education system also allows learners to study Life Orientation as a subject. The subject is based on the understanding of the constitutional rights of South African citizens and the promotion of human health, physical education, and environmental responsibilities.

The blend of different subjects, as noted by the interviewed children and through observations, made them active in conservation of the environment, as they were involved in tree planting with the help of their teachers and other non-teaching staff in the schools. Teaching learners the Afrikaans and Setswana languages helped in disseminating education to learners in the local language that they can understand better, based on the fact that the Tshwane North District contains a large percentage of poor and middle-income earners who converse in the various local languages. Afrikaans and Setswana are some of the home languages spoken in the region. It is therefore easier to explain to these students critical ideas such as environmental conservation efforts, because they understand the local languages better than the English language.

The teachers interviewed also claimed that they used home languages to explain issues and concepts related to conservation efforts. By explaining issues in home languages, the learners can gather a better understanding of what is expected from them, especially when it comes to matters of conserving the environment. Hence, one teacher said: “*our learners do not use English at home. They speak their different languages. As a concerned teacher I use their home language to teach them EE issues. This strategy has positive effect. I know that they will tell their parents about EE and the message will spread to the entire community.*”

The non-teaching staff were also interviewed. They pointed out that, although they do not know the subject that is taught to learners, they noticed that learners are conscious of the cleanliness of their environment. As for the non-teaching staff, they said that they contribute towards the conservation of the environment supporting teachers in cautioning some of the learners who throw papers around and encourage them to pick them. They declared that “*we all encourage children in participating on conservation efforts within and out of the school environment*”. All education stakeholders are therefore important in EE provision and practice.

### 4.2. Sources of Environmental Education

Teaching needs to be complemented with available learning materials so that learners can make references and build their knowledge and understanding on different issues relating to the environment. There is limited internet coverage in the region, and the findings noted that internet was a less popular as a source for EE in the Tshwane North District region. This could be attributed to the fact that, at the primary school level, children have limited exposure, as there is limited internet infrastructure in the schools compared to high school and tertiary learning institutions. Limited access or lack of technology is depriving learners an opportunity to learn through modern models. One teacher from a poverty-stricken area said: “*there are videos that we can use to enhance teaching and learning. But unfortunately our school does not have computers, wifi or data. Even if I try to use my phone not all learners can see. The screen is very small. We need these resources to promote EE*”.

This has made the internet less popular among children in primary schools engaged in the research. The information sources may be out of formal classes, but they are accessible from within the school environment and outside of the school, including the surrounding community where the learners live. Moreover, as noted by Altin et al. (2014) [26] in a study conducted in Turkey, the mass media has been used as one of the key contributors towards the provision of EE awareness to learners in primary and secondary schools. This is also the case in the Tshwane North District, where local media outlets, such as *Grootfm90.5*, *Tshwane FM*, *Moretele Community Radio*, *Radio Pulpit*, and *Family Radio*, are among the various radio channels frequented locally. These radio stations contributed to sensitizing the community, including the learners, on environmental insights that include the need to conserve the environment and how learners can achieve it for the betterment of the society to create a sustainable universe. With regard to formal learning in schools, the research noted that the books and other publications available in the school library helped to provide EE to learners, by complementing the knowledge that teachers provided to learners in class and practical lessons.

### 4.3. Awareness of Environment Problems

It is important to, first of all, determine whether the learners in these schools are aware of the environmental problems that exist in society. This is essential as the first building block in determining the success of teaching programs used in the region, and whether they are effective in providing learners with the relevant knowledge on conservation efforts.

Being aware of environmental challenges is a step towards providing EE and introduce remedial measures to address the negative impact, as noted by the teachers and support staff. It is evident that primary schools in the region contribute enormously towards teaching learners the importance of conserving the environment, and what it takes to ensure that there is adequate awareness of the need to conserve and reduce problems associated with environmental degradation.

When tasked to respond on how they are able to acquire the vast knowledge on environmental problems, learners claimed that school events, such as environmental clean-up days, tree planting campaigns, 4K clubs, and participation in national events such as charity foundation events, greatly contributed to shape their perception of conserving the environment and what it takes to reduce problems associated with poor environment conservation. Regarding awareness campaigns held in schools, one learner said: “*what I like about these events is that there is drama in which learners act on issues of the environment, some learners participate in dance groups, and they play the music I like*”. An observation made within the schools noted events such as tree nurseries and EE charts that were hung in various classes that the researcher visited. This is an indication that learners have adequate access to environmental knowledge, which in turn they use to make informed decisions when it comes to conservation efforts.

### 4.4. Environmental Education Administration in Schools

The school environment can testify whether the institutions are providing environmental conservation education to learners, based on observations of how the school looks. The schools that we visited during the research had a number of activities, such as nurseries of endangered tree plant species, including the eucalyptus and the blue gum in schools such as Baxoxele Primary School, Dilopye Primary School, and Busy Bee Junior Primary School. A teacher from one of these schools said: “*we met as teachers teaching EE in the three schools and identified indigenous trees that are disappearing in the landscape. We collected seeds and planted them in the school yard. Every time we talk about endangered trees we show learners these trees from our school garden. We hope that they will discuss with their parents about cutting trees, especially endangered ones*”. These spectacular scenes with abundant trees are a clear manifestation that the current curriculum in the region advocates and enable education stakeholders to emphasize conservation efforts and ensure that learners have adequate knowledge and skills in EE.

External environmental issues, such as vandalism of school property and the level of cleanliness, were also assessed, as it is part and parcel of conservation and protection efforts. The research noted a number of broken school desks and writing on the walls. Other schools were cleaner, which could be an indication that, despite the provision of EE to learners, the level of enforcement of the policies and the knowledge provided differ among schools in the region. Therefore, there is need for stakeholders, such as teachers and non-teaching staff, to enforce the implementation of these programs after teaching in class, so that learners are aware of conservation and adequately practice it on the school grounds.

## 5. Discussion

The responsibility of managing the ecosystem is entirely entrusted to human beings and, therefore, there is a need for every individual to contribute towards sustainable environment development, and this duty should be supported by learning institutions that act as academic provision centres. Evidence of EE provision was available in the schools that the research focused on, although the level of commitment was not as evident during observation. This is because there was disparity in performance and demonstration of application of the EE in these schools, which is attributed to the differences in topography and managerial systems that are being implemented by each head teacher in their administration jurisdictions (Gopalan, 1999) [27].

Education on conservation cannot only be provided in education centers but should also be provided in the society where people live and operate from. This implies that the government and conservation non-governmental organizations can embrace different mechanisms, such as using public gathering sites including parks and beaches, to erect signposts to sensitise the public on conservation importance (Mtaita, 2007) [28]. Moreover, mass media outlets are also playing a vital role in creating awareness of the importance of conservation in society. The research findings noted robust activities within the region that act as conservation activities. Moreover, the use of print media and mass media within the region to sensitise the general public on conservation efforts are not only aimed at creating awareness with the people, but they are also helping school children to obtain complementary knowledge on conservation efforts that they should emulate and embrace. This will make it possible for the region to experience adverse climate change by bringing everybody on board to engage in conservation activities and reduce negative environmental degradation attempts.

On the other hand, the attitude and the behaviour of learners is critical in determining the impact EE create in their lives, and what approach the teachers use when they educate learners on EE programmes. Attitude and behaviour shape one’s interest in studying a certain concept, and also directly impacts on how they will emulate and embrace the knowledge provided for future use (Erhabor and Don, 2016) [29]. The research noted that learners were interested to learn subject concepts that are related to conservation efforts, not only because it is part of the curriculum they must learn to pass, but also because the curriculum related to real-life events happening on a daily basis. This idea of embracing conservation efforts show that learners are determined to learn and contribute to conservation efforts through the knowledge provided. Hence, they showcased a positive and natural embracing attitude that welcomed any effort to conserve the environment.

## 6. Conclusions

The curriculum has offered an opportunity for the learners to learn and embrace conservation activities from an early stage of development in their lives. Primary school, as the first point of entry for learners to the academic world, provides an opportunity for stakeholders in the education sector to design the teaching curriculum in a manner that ensures that EE is incorporated in the program, as it is the case with the South African primary school curriculum. We established that the curriculum offers opportunities to learners to study EE and broadens their consciousness about the environment. We also observed teachers’ teaching time in different subjects to impart knowledge about EE. What stands out in the way teachers impart EE is that they integrate academic work with cultural activities. However, we also observed that teachers and learners experience challenges in the teaching and learning of EE. Their efforts are hampered by a lack of resources and technologies for use in the classroom. Despite these challenges, teachers and learners persist with the help of the community.

The study, however, proposes a number of recommendations that should be embraced by stakeholders to enhance and improve EE provision in the region.
As the education sector is the largest government unit that is accessible across the various regions, these institutions must be empowered to focus more on conservation and ensure that learners are adequately sensitised to participate in conservation efficiently.The community must be brought on board along with sustainability matters, where the current awareness campaign should be extensively carried out rather than only using signposts to inform them. This implies that public rallies similar to election campaigns must be deployed during the period of conservation campaigns.Enhanced access to proper sanitation and clean water should be prioritised by government and conservation activists. Focusing on one aspect at a time within the scope of environmental sustainability promotion will result in a positive improvement of the society.

## Data Availability

The data presented in this study are available on request from the corresponding author. The data are not publicly available due to confidentiality and anonymity clauses in the study.

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
