# Peer review of "Learners’ Knowledge of Environmental Education in Selected Primary Schools of the Tshwane North District, Gauteng Province"

_ijerph, 2022, doi:10.3390/ijerph192315552_

Round 1

Reviewer 1 Report

This article deals with a very interesting and relevant topic. The theoretical framework is quite well developed, perhaps even a little extensive, taking into account the size of the paper. However, the results need further elaboration and detail in the description. For example, how many students were actually interviewed? Surely it wasn't the 80 that are mentioned. And how many staff members? What were the objectives of each type of interview? On the other hand, evidence from these interviews is not presented. Excerpts from the interviews are never presented, illustrating what is mentioned by the authors, which in itself constitutes already an interpretation. The authors present their own inferences, without actually presenting the perspective of the participants themselves. Finally, the final discussion does not appear to be evidence-based. For example, the final recommendations do not appear to be supported by the data collected (they could be made even without data collection). Other examples: it is recommended by the authors that: the use of print media and mass media within the region to sensitize the general public on conservation efforts are not only aimed at creating awareness with the people, but they are also helping school children to get complementary knowledge on conservation efforts that they should emulate and embrace – where is the evidence? This is a conclusion of the authors based on what data?  or in the statement: The research noted that learners were interested to learn subject concepts that are related to conservation efforts, not only because it is part of the curriculum they must learn to pass, but also because the curriculum related to real life events happening on a daily basis – once again, what is the evidence gathered in this investigation that supports this statement? In summary, the results and discussion section needs some reformulation and deepening.

Author Response

Methodology - sample number included

Results - detailed words of participants

Conclusion - revised to cover essential aspects of aims and findings 

Reviewer 2 Report

The problem in this research relating to the interactions between teachers and students is significant in the context of environmental issues. The theoretical framework is well developed. However, several underlined references are not in the reference section (see below). Qualitative methodology is appropriate for this type of research. On the other hand, more details about participants and their numbers are required for the results' validity. Also, it would be essential to present in a synthesis table the results of the analysis of the data of the three groups: teachers (n = 10), non-teaching staff (n = XX), and learners (n = 80)]. Regarding the non-teaching staff, their number is not indicated.

The following works are not listed in the references:

Gough (2006)

Paw and Yaday (2017)

Rezaei et al., 2016

Kals et al., 1999

NRC, 2007

Kamar et al., 2015

Mjimba (2020)

Van and 259 Dunlap (1981)

DoE (2021)

The following references are not cited in the text:

Van Koppen, B. and Schreiner, B., 2014.

Yachina, N.P., Khuziakhmetov, A.N. and Gabdrakhmanova, R.G., 2018.

Author Response

Methodology - sample number included

Results -  detailed words of the participants

Conclusion - revised to cover essential aspects of aims and findings

Omitted references - now included

Uncited references - removed 

Round 2

Reviewer 1 Report

The results are now more complete and based on the evidence obtained. Although the discussion could still be improved towards the results obtained, in general I think the paper is in a position to be accepted.